# Discovery of a weak topological insulating state and van Hove singularity in triclinic RhBi$_2$

Kyungchan Lee[1,2], Gunnar F. Lange[3], Lin-Lin Wang [1,2], Brinda Kuthanazhi[1,2], Thaís V. Trevisan [1,2], Na Hyun Jo[1,2], Benjamin Schrunk[1,2], Peter P. Orth [1,2], Robert-Jan Slager [3,4✉], Paul C. Canfield[1,2✉] & Adam Kaminski [1,2✉]

Time reversal symmetric (TRS) invariant topological insulators (TIs) fullfil a paradigmatic role in the field of topological materials, standing at the origin of its development. Apart from TRS protected strong TIs, it was realized early on that more confounding weak topological insulators (WTI) exist. WTIs depend on translational symmetry and exhibit topological surface states only in certain directions making it significantly more difficult to match the experimental success of strong TIs. We here report on the discovery of a WTI state in RhBi$_2$ that belongs to the optimal space group P$\bar{1}$, which is the only space group where symmetry indicated eigenvalues enumerate all possible invariants due to absence of additional constraining crystalline symmetries. Our ARPES, DFT calculations, and effective model reveal topological surface states with saddle points that are located in the vicinity of a Dirac point resulting in a van Hove singularity (VHS) along the (100) direction close to the Fermi energy ($E_F$). Due to the combination of exotic features, this material offers great potential as a material platform for novel quantum effects.

[1] Ames Laboratory, Ames, IA, USA. [2] Department of Physics and Astronomy, Iowa State University, Ames, IA, USA. [3] TCM Group, Cavendish Laboratory, University of Cambridge, Cambridge, UK. [4] Department of Physics, Harvard University, Cambridge, MA, USA. ✉email: rjs269@cam.ac.uk; canfield@ameslab.gov; adamkam@ameslab.gov

From the perspective of considering solely TRS, 3D topological insulators are classified by a series of $\mathbf{Z}_2$ topological invariants, $(\nu_0;\nu_1\nu_2\nu_3)$. The case $\nu_0 = 1$ is referred to as a strong topological insulator (STI) [1,2]. A characteristic feature of a STI is the existence of gapless topological surface state (TSS) on all surfaces, which have been observed experimentally in many materials such as $Bi_{1-x}Sb_x$, $Bi_2Se_3$, and $Bi_2Te_3$ [3]. In contrast to 2D TIs, even when $\nu_0 = 0$ the system can have non-trivial $\nu_i$, resulting in a weak topological insulator (WTI). Reported WTI states are rather rare as gapless TSS can be detected only on particular surfaces, which may not be natural cleaving planes in many 3D materials. For instance, $Bi_{14}Rh_3I_9$ was predicted to be a WTI with $\mathbf{Z}_2 = (0;001)$. However, direct experimental verification of the TSS has remained elusive since the cleaving plane is (001), which is a topologically trivial side [4] and hence forces reverting to indirect signals on e.g., step edges[5]. The recent development of nano-ARPES has reinvigorated the search for WTIs. For instance, a first WTI TSS was only recently measured by isolating a ~2 μm size side facet and using nano-ARPES in quasi-one-dimensional bismuth iodide $\beta$-$Bi_4I_4$[6].

As part of an effort to design and discover materials with novel, intrinsically non-trivial, topological properties we identified $RhBi_2$ as an exciting candidate material. To minimize potential degeneracies in reciprocal space, we decided to start with the lowest possible symmetry unit cell: triclinic. To maximize spin-orbit coupling we chose the heaviest (essentially) stable element, Bi as a majority component. For simplicity we chose to start by limiting our search to binary compounds. As a result of this cascade of physically inspired constraints we rapidly identified $RhBi_2$ as a promising candidate material with clear van der Waals-like bonding along the crystallographic $a$-axis (Fig. 1b). $RhBi_2$ grows in triclinic, $RhBi_2$ structure (aP12, space group 2). $RhBi_2$ is the only known compound to this specific, very low symmetry, structure. Here, we provide theoretical understanding and experimental evidence of the simplest WTI state in triclinic crystal $RhBi_2$ with clear TSS at the natural cleaving surface. As we will discuss below, the WTI classification of $RhBi_2$ maximally profits from its space group. As $P\bar{1}$ only features inversion in addition to translations, all topological properties in the presence of TRS can faithfully be determined from the parity eigenvalues of the occupied bands at TRIM points[7]. Moreover, due to the absence of additional symmetries none of these high symmetry momenta are related and a WTI can exist in every stacking direction[8]. In fact, this carefully selected, simple structure makes this the only space group in which the symmetry eigenvalues at TRIM points convey all topological invariants one-to-one[9–12]. Rather surprisingly, we find that this freedom of choosing the stacking direction culminates in an index, (0;001), that is perpendicular to the layering of the material, creating the possibility of a TSS preserving cleavage plane.

The lack of symmetries, especially the absence of rotational symmetry, in triclinic $RhBi_2$ allows the system to have a saddle points, where the curvature of a band has opposite sign in two perpendicular direction in momentum space. A saddle point is closely related to a VHS, leading to a divergence in the density of states. Previously, VHSs have been reported in a variety of materials; graphene[13,14], cuprate superconductors[15–17], $Pt_2HgSe_3$ [18], $Pb_{1-x}Sn_xSe(Te)$[19]. The proximity of $E_F$ to a VHS is expected to significantly enhance the impact of electronic correlations and drive quantum many body instabilities toward superconductivity[20–22] ferromagnetism[23–26] or antiferromagnetism[27], depending on band structure and many-body interactions. In addition, the subtle interplay of protected and correlated surface states with the bulk is predicted to lead to unusual surface quantum criticality[28]. As we show, this is expected to result in an order of magnitude enhancement of superconducting $T_c$, and generally increases the

tendency toward development of electronic order. Partial substitution of Rh with Ru or electrostatic gating offer a possible path for lowering the $E_F$ in $RhBi_2$. Owing to the simplest topological classification and VHS in the vicinity of $E_F$, triclinic crystal $RhBi_2$ offers unique advantages for exploring exotic quantum phenomena in a weak topological insulator.

We present a study of triclinic $RhBi_2$ by using ARPES, DFT calculations and an effective model. Our result reveals two surface Dirac points, which are protected by time-reversal symmetry at the $\bar{\Gamma}$ and $\bar{Z}$ points. We focused experimental attention at the momentum space around the $\bar{Z}$ point of the surface BZ. The shape of the Dirac cone around $\bar{Z}$ is significantly modified from typical Dirac dispersion due to the presence of two saddle points associated with a VHS. The saddle points are located at a binding energy of ~ 80 meV. Based on DFT calculations, we identify that surface state of $RhBi_2$ along (100) has a WTI phase, which is characterized by $\mathbf{Z}_2 = (0;001)$.

## Results

**Band structure calculations.** Due to the symmetry group hosting only an inversion center, the layered triclinic $RhBi_2$ is an interesting case in terms of both band structure and topology. The bulk band structure of $RhBi_2$ calculated from density functional theory[29,30] (DFT) using PBE exchange correlation functional with spin-orbit coupling (SOC) is plotted in Fig. 1c. The 3D FS at the $E_F$ is shown in Fig. 1e with the eight time-reversal invariant momentum (TRIM) points and reciprocal lattice vectors labeled. By virtue of inversion symmetry and TRS, all bands are doubly degenerate in the BZ due to Kramers theorem. The bands of interest arise by hybridization from Rh $d$ and Bi $p$ orbitals. From the small but finite density of states (DOS) at $E_F$ in Fig. 1d, we see that $RhBi_2$ is a metal exhibiting non-overlapping pockets. There are no crossings between the top valence and bottom conduction bands anywhere in the BZ. The two bands crossing the $E_F$ give multiple electron and hole pockets. The electron and hole pockets are near BZ boundaries, while there is a large band gap around the BZ center Γ point. Notable are the electron pockets around X and near L points and the hole pockets near the Y point. There are also two small hole pockets totally inside the BZ. Although the structure is stacked on the bc plane along the $b_1$ direction, a large electron pocket is on the $b_1 = 0.5$ plane around the X point. In contrast, there is band gap on the $b_3 = 0.5$ plane around the Z point.

Topological analysis. With inversion being the only symmetry in the system besides time-reversal symmetry, the topological features are determined by the parity eigenvalues of the occupied bands at TRIM points[7]. In fact, from the perspective of the recent advancements in classifying band structures using constraints in symmetry eigenvalues[10] and comparing them to atomic configurations[11,12], the space group at hand, number 2 or $P\bar{1}$, is the only one in which such symmetry indicators unambiguously define all topological invariants[8,9], as there are no symmetry constraints between high symmetry momenta[8]. The topological indices, apart from TRS invariant $\nu_0$ that we find to be zero, comprise three $\mathbf{Z}_2$ invariants corresponding to the weak indices and a $\mathbf{Z}_4$ index[31] that comes from promoting $\nu_0$ to the inversion symmetric context. The latter corresponds to the twice the value of the inversion invariant $\delta_i$, $\mathbf{Z}_4 = 2\delta_i$, here[9]. The calculated $\mathbf{Z}_4$ index is 2, indicating that $RhBi_2$ is topologically non-trivial, but indeed not a strong topological insulator. The calculated parity eigenvalue products are (+1) at Γ and Z points, and (-1) at all the other TRIMs, which gives the Fu-Kane index of (0;001), determining the three $\mathbf{Z}_2$ invariants. This can also be confirmed by Wannier charge center (WCC) evolution or Wilson loop on $k_3 = 0.0$ and 0.5 planes in Fig. 1g and g, showing 2D $\mathbf{Z}_2 = 1$ on these planes[32–34]. In contrast, 2D $\mathbf{Z}_2 = 0$ on the other $k_{1,2} = 0.0$ or

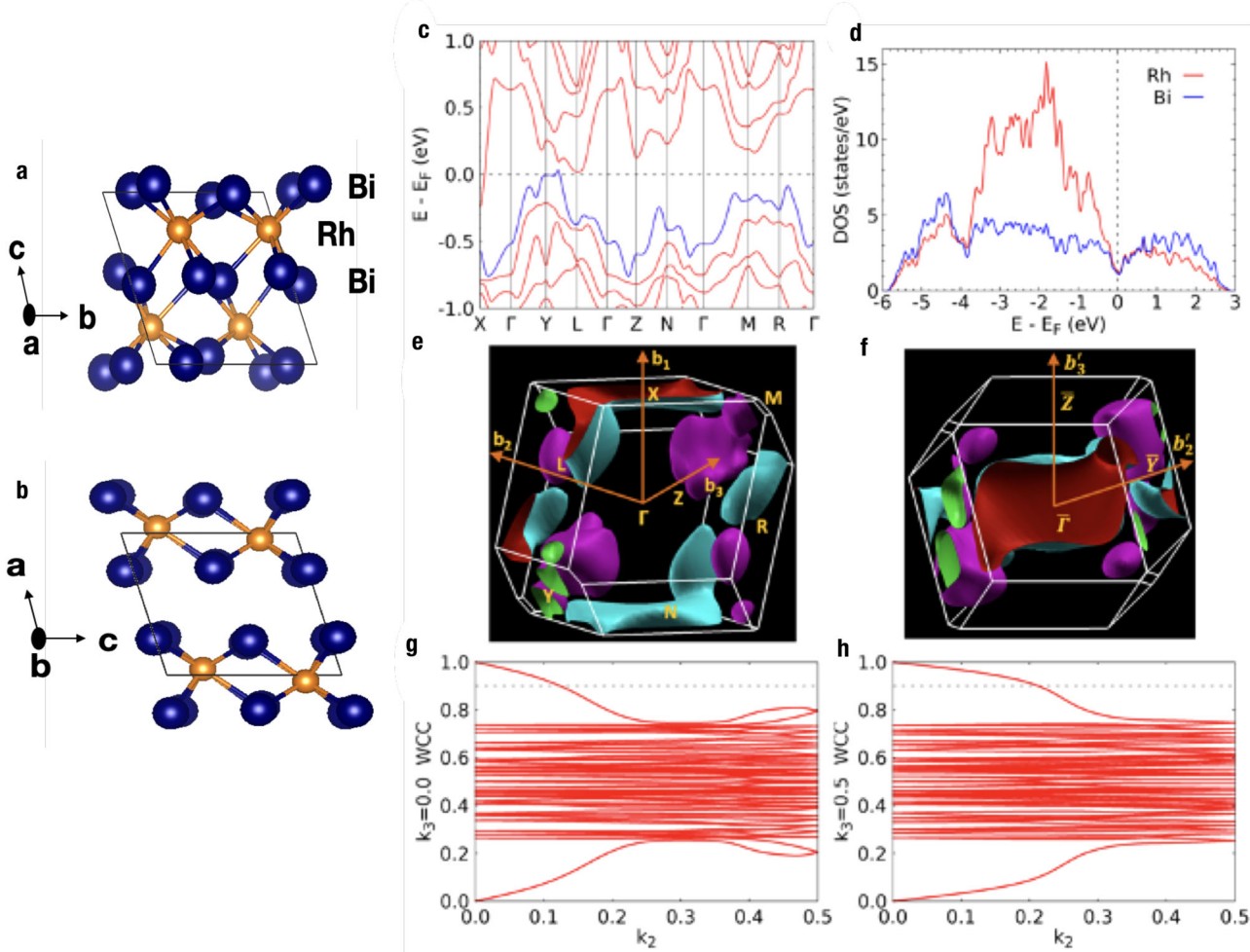

**Fig. 1 Calculated bulk band structure, DOS and WCC evolution of RhBi$_2$. a-b** Crystal structure of triclinic RhBi$_2$ with different orientations (Bi:blue spheres, Rh: yellow spheres). **c** Bulk band structure of RhBi$_2$ in the triclinic crystal structure (space group 2) calculated in DFT-PBE with SOC. The top valence band according to electron filling is in blue. **d** Density of states (DOS) projected on Rh and Bi. **e** 3D Fermi surface at the ($E_F$) with eight time-reversal invariant momentum (TRIM) points and reciprocal lattice vectors labeled. Electron and hole pockets are in cyan (red inside) and purple (green inside) respectively. **f** 3D FS viewed along b$_1$ axis with projection of TRIM points and 2D reciprocal lattice vectors labeled. **g** and **h** Wannier charge center (WCC) evolution on the k$_3$=0.0 and 0.5 planes, respectively, showing the non-trivial topology.

0.5 planes. Interestingly, although the layered structure is stacked along the b$_1$ direction, the weak TI can be seen as a stacking of quantum spin Hall (QSH) layers along the b$_3$ direction.

In Fig. 1f, the 3D FS is rotated for the view along b$_1$ or perpendicular to the *bc* plane, i.e., (100). On the (100) surface, the $\Gamma$ and X points are projected to $\bar{\Gamma}$ point, Z and N to $\bar{Z}$ point, and Y and L to $\bar{Y}$ point, respectively. This shows that for the 2D FS on (100) around the $E_F$, the area near the $\bar{Z}$ point is gapped without bulk band projection, while the $\bar{\Gamma}$ and $\bar{Y}$ points are surrounded by bulk states. For bulk-boundary correspondence of TIs[35], parity eigenvalue products can also be projected onto surface as in $\pi_a = \delta_{a1}\delta_{a2}$, where TRIM a$_1$ and a$_2$ projected into a on surface. The result is ($-1$) for $\bar{\Gamma}$ and $\bar{Z}$ points, while ($+1$) for the others. Thus, an even number of two surface Dirac points will emerge on (100), one at $\bar{\Gamma}$ and the other at $\bar{Z}$. In summary, RhBi$_2$ is a topologically non-trivial 2D material as a layered compensating semimetal hosting the simplest WTI possible as the underlying space group has minimal symmetry.

**Electronic structure measured by ARPES.** In order to verify the nontrivial topological nature of triclinic RhBi$_2$, we performed ARPES measurements and DFT calculations to identify relevant

features in the electronic structure. Fig. 2a shows projection of TRIM points and the FS of triclinic RhBi$_2$ measured using laser-based APRES, where dashed lines mark main crystallographic directions in the BZ. The experimental data in Fig. 2a may appear to have twofold symmetry, but upon closer inspection it does not. The left and right surrounding of the $\bar{Z}$ pocket are quite different, as there is additional FS sheet merging with bulk bands to the right of Z pocket (marked by red circle in Fig 2a), but it is absent on the left side. This is in good agreement with calculations in Fig. 2d, where the sharper surface state is present on the right side of Z pocket, but no on the left. To determine projection of TRIM points on FS, we analyze Momentum distribution curves (MDCs) and Energy distribution curves (EDCs), Dashed lines are marked along two different projection of TRIM points ($\bar{\Gamma}$-$\bar{Y}$ and $\bar{\Gamma}$-$\bar{Z}$). Since It is well known that surface Dirac points appear at TRIM points, we used location of surface Dirac points to determine TRIM points. We carefully checked MDCs and EDCs to verify the location of $\bar{Z}$ point. Fig. 2h shows MDC extracted at chemical potential. Fig. 2i shows EDCs at centers of two MDC peaks around $\bar{\Gamma}$ and $\bar{Z}$, respectively. The measured band dispersion along those lines are shown in Fig. 2b and c. Two Dirac points are clearly present in the data: one located at the $\bar{\Gamma}$ point with a

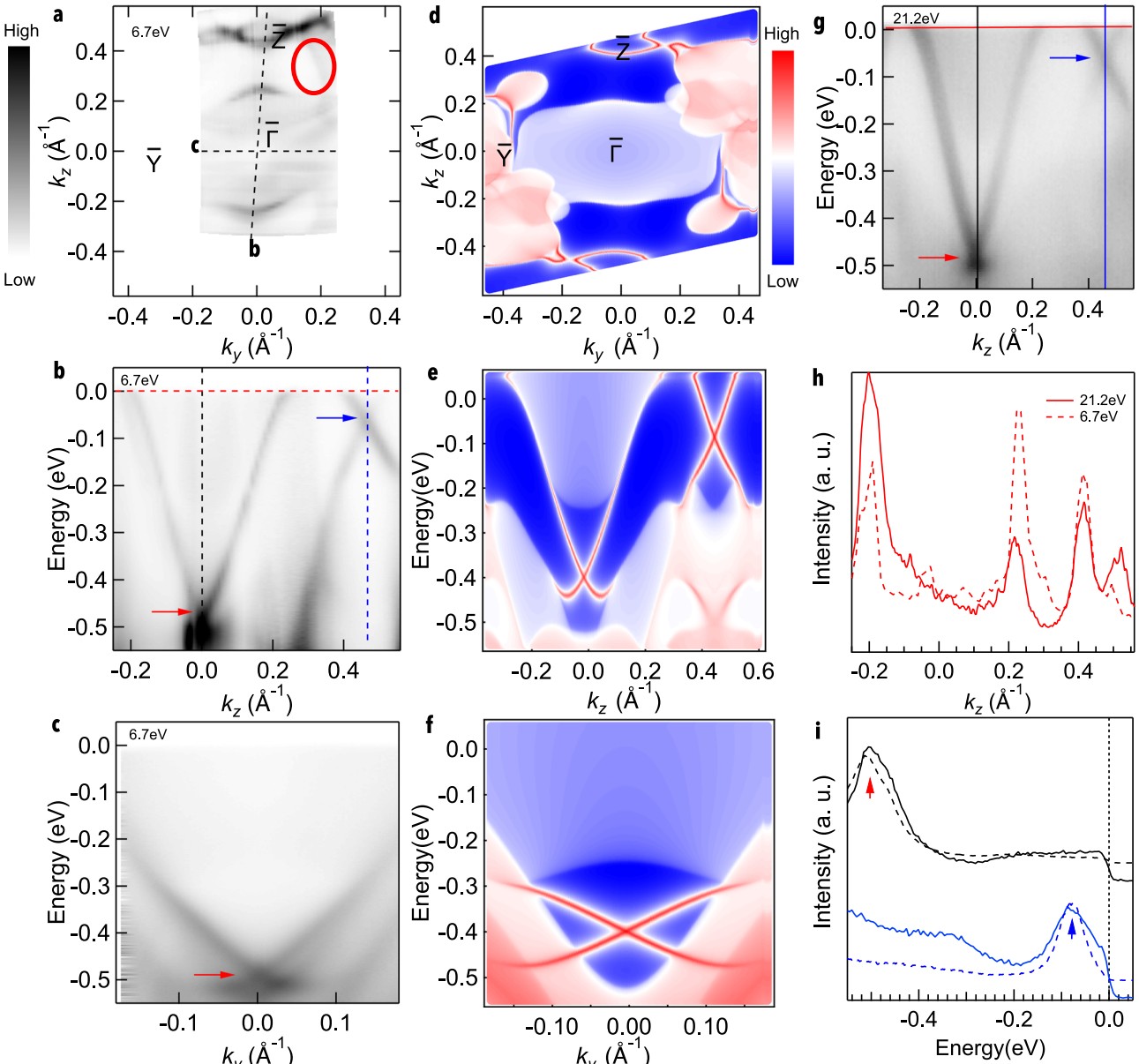

**Fig. 2 Experimental and calculated Fermi surface and band dispersion of RhBi$_2$ at $T = 40$ K. a** Fermi surface plot of the ARPES intensity integrated within 10 meV of the chemical potential and measured with photon polarization along $\bar{\Gamma}$-$\bar{Z}$ direction. **b** Band dispersion along the $\bar{\Gamma}$-$\bar{Z}$ line. **c** Band dispersion along $\bar{\Gamma}$-$\bar{Y}$. **d** DFT calculations of Fermi surface. **e**–**f** DFT calculations of energy dispersion corresponding to the ARPES data (**b**) and (**c**), respectively. **g** band dispersion along $\bar{\Gamma}$-$\bar{Z}$ measured using photon energy of 21.2eV, unpolarized. **h** Momentum distribution curves (MDCs) at the chemical potential at photon energy 6.7 ev and 21.2 eV. **i** Energy distribution curves (EDCs) at the $\bar{\Gamma}$ and $\bar{Z}$ in two different photon energy. Red arrow marks position of the surface Dirac point at $\bar{\Gamma}$ ($k_z = 0$ Å$^{-1}$) and blue one marks the position at the $\bar{Z}$ ($k_z = 4.5$ Å$^{-1}$). Dashed lines are extracted at a photon energy 6.7 eV and sloid ones are extracted at 21.2 eV.

binding energy of ~ 500 meV and the other at the $\bar{Z}$ point with a binding energy of ~80 meV. The red arrow marks the Dirac point at the $\bar{\Gamma}$ point and the blue arrow marks the Dirac point at the $\bar{Z}$ point. Fig. 2c shows the band dispersion along the $\bar{\Gamma}$-$\bar{Y}$ direction where the red arrow marks the location of the Dirac point there. Fig. 2d–f shows DFT calculations along the same directions as for the ARPES data. Overall, DFT calculations agree quite well with experimental data. Unlike typical TSS of STIs, which have a single Dirac point, we observe two surface Dirac points at TRIM points. This apparent discrepancy between STIs and WTIs shows that triclinic RhBi$_2$ possesses TSS arising from weak topological insulating phase in (100) direction. In addition, we observed twofold rotational symmetry at the FS from the DFT calculations,

although the crystal structure is triclinic. This is because three-dimensional inversion symmetry reduces to effectively two-dimensional rotation symmetry upon projection of all transverse momenta. To understand this phenomenon, we carefully studied different projections of each layer in the unit cell. Supplementary Note 3 shows different projections of $k_1(k_x)$ values on (100) surface. Red looped curves represent electron pockets. Based on the calculation, only projections from $k_1 = 0$ and 0.5 planes show twofold rotational symmetry and the $k_1 = 0$ plane shows large electron pocket around the center of the BZ. Therefore, the large electron pocket around $\bar{\Gamma}$ is mainly from the $k_1 = 0$ plane and addition of all planes in the BZ will give us the effective twofold rotational symmetry on the 2D FS.

To demonstrate surface state origin of the measured band dispersion, we plot data along a $\bar{\Gamma} - \bar{Z}$ direction that were measured using different photon energy (21.2 eV) in Fig. 2g. The sharp band dispersion measured using 21.2 eV photons is identical to the one measured using 6.7 eV and shown in Fig. 2a, which proves its surface state origin. To demonstrate this better, we plot MDC at the chemical potential in Fig. 2h and EDCs at the two Dirac points in Fig. 2i for the two photon energies. Although there is some variation in overall intensity due to matrix elements, the peaks in solid curves measured using 21.2 eV and dashed curves that were measured using 6.7 eV occur at the same momenta/energies which demonstrates lack of dispersion along direction perpendicular to sample surface and thus surface state origin. We also note that while 6.7 eV data was measured using photons polarized along $\bar{\Gamma} - \bar{Z}$ direction, the 21.2 eV data was measured with unpolarized beam, thus there are no strong polarization effects.

**Saddle points and TSSs around the $\bar{Z}$ point**. Out of the two surface Dirac points present, the Dirac point around the $\bar{Z}$ point is by far the more interesting. Instead of a typical Dirac like dispersion[36], it shows a strong warping effect along the energy axis making saddle points in the lower band. $k \cdot p$ model band dispersion in Fig. 3a illustrates the Dirac point and two saddle points that exist in proximity of the $\bar{Z}$ point. The FS in this area of BZ is plotted in Fig. 3b. Fig. 3c-f displays calculated DFT band dispersion in the vertical direction(#1-3) and one horizontal cut (#4) which are marked as dashed lines in Fig. 3b. Interestingly, the vertical cuts (along $k_z$), Fig. 3c and e shows the lower band with negative curvature with $k_z \sim 0.45$ Å$^{-1}$ while the horizontal cut (along $k_y$), Fig. 3f, shows the lower band with positive curvature. Along cut 4, there are actually two upper bands that give rise to two local minimums in energy as marked by black arrows. Those two minimums give rise to two saddle points as demonstrated in Fig. 3a. In cut 1 we can see the upper and lower bands are separated in energy (Fig. 3c). Band dispersion along the $\bar{Z}$-$\bar{\Gamma}$ line is shown in Fig. 3d. Dirac dispersion is clearly visible with Dirac point at the binding energy of ~80 meV and is marked by the black arrow. Figure 3e displays the band dispersion at the cut 3, where the two bands are clearly separated as in cut 1. The separation in Fig. 3c and e is fairly symmetric with respect to the Dirac point. Corresponding ARPES data, measured along the same cuts in the BZ are shown in Fig. 3g-j. Based on calculations and the experimental result, we conclude that this is the surface state forms a Dirac point rather than a line of Dirac dispersion along the $k_y$ direction. On the other hand, band dispersion in the horizontal direction significantly deviates from the vertical directions as shown in Fig. 3f and j. Both upper and lower bands have a parabolic dispersion instead of the linear dispersion. Naturally, the lower band shows positive curvature. As a result, it looks like the overlap of two parabolic bands with slightly different curvatures.

To qualitatively address the observed saddle points in vicinity of the $\bar{Z}$ point on the (100) surface we consider a simple effective model. As we have time-reversal ($\mathcal{T}$) and inversion symmetry ($\mathcal{P}$) in the bulk, $\mathcal{A} = \mathcal{P}\mathcal{T}$ commutes with Hamiltonian, reflecting the Kramers degeneracy by virtue of squaring to $-1$. The presence of this symmetry will constrain the form of the bulk Hamiltonian affecting the effective model for the surface after projecting. This will be reported elsewhere and we simply note that $\mathcal{P}$ is formally broken on the surface. In fact, having the inversion center at the origin ensures that surface states with opposite $\mathcal{P}$-value reside on opposite surfaces. Hence, as motivated in Supplementary Note 2, we consider a simple two-band model that is only constrained by

TRS to describe the strongly anisotropic anomalous edge states,

$$H(\mathbf{k}) = d_0(\mathbf{k})I + d_1(\mathbf{k})\sigma_x + d_2(\mathbf{k})\sigma_y + d_3(\mathbf{k})\sigma_z \quad (1)$$

In the above $d_0(\mathbf{k})$ is even in $\mathbf{k}$ and $d_{1,2,3}(\mathbf{k})$ are odd in $\mathbf{k}$. Expanding our energy to second order then gives

$$\varepsilon_\pm(\mathbf{k}) = E_0 + A_1 k_y^2 + A_2 k_z^2 + A_3 k_y k_z \pm \sqrt{\sum_{i=1}^{3}\left(B_i k_y + C_i k_z\right)^2}, \quad (2)$$

where $E_0$ is the energy at the crossing points of the surface bands, $\{A_i\}$, $\{B_i\}$ and $\{C_i\}$ are all real parameters and momenta are measured relative to the $\bar{Z}$ point. To accommodate strain effects, we could also expand $H(\mathbf{k})$ to third order, giving:

$$\begin{aligned}\varepsilon_{3\pm}(\mathbf{k}) = E_0 &+ A_1 k_y^2 + A_2 k_z^2 + A_3 k_y k_z \\ &\pm \sqrt{\sum_{i=1}^{3}\left(B_i k_y + C_i k_z + D_i k_y^2 k_z + E_i k_y k_z^2\right)^2}\end{aligned} \quad (3)$$

We use simulated annealing to fit this model to our surface band structure around the $\bar{Z}$ point. This gives the parameters shown in table I and II in Supplementary Note 2. Our parameter space is very high dimensional, so we cannot hope to find the physically relevant parameters. This will be rectified in a future work. As we here are interested only in an effective model to compute the DOS, and as we have the full DFT results available, any model which reproduces the energy dispersion close to the crossing point is sufficient.

**van Hove singularity and divergence in the surface state DOS**. Extrema of the energy dispersion, $\nabla \varepsilon_\mathbf{k} = 0$, are called van Hove points and lead to singularities in the density of states (DOS): $g(E) = g_s \int \frac{d^{d-1}k}{(2\pi)^d} \frac{1}{|\nabla \varepsilon_\mathbf{k}|_{\varepsilon_\mathbf{k}=E}}$ ($g_s$ denotes spin degeneracy). In two dimensions, $d = 2$, saddle points of the dispersion lead to a logarithmically divergent DOS. If this divergence occurs close to the Fermi energy, electronic correlations are strongly amplified, which can drive the system toward various electronically ordered states such as superconductivity, spin or charge density waves. In contrast, nodal points of the dispersion, where the Fermi surface shrinks to a point are characterized by a vanishing DOS.

As described above, our ARPES measurements indicate the presence of both van Hove and nodal points in the surface bands of RhBi$_2$ that occur in close proximity to each other. To further underpin this observation, we here study in detail the properties of a low-energy $k \cdot p$ model (see Eq. ((2))) that describes the band structure around the $\bar{Z}$ point in the surface BZ (sBZ). As shown in Fig. 3a, the surface band structure exhibits a nodal point at the $\bar{Z}$ point, $(k_{\bar{Z},y}, k_{\bar{Z},z}) = (0, 0.5)\frac{\pi}{a}$. At this point the singly degenerate lower and upper bands touch at an energy $E = -82$ meV below the Fermi energy $E_F = 0$. Both bands are anisotropic away from the nodal point due to the low symmetry of the material. This is particularly noticeable in the lower band, where the dispersion along the direction $k_y - k_{\bar{Z},y} = -4.15(k_z - k_{\bar{Z},z})$ is almost flat and quadratic, while it is much steeper in the orthogonal direction (see Fig. S4 in the SI). Importantly, the lower band features two saddle points near $\bar{Z}$ at $(k_y, k_z) \approx (-0.042, 0.510)\frac{\pi}{a}$ and $(0.042, 0.490)\frac{\pi}{a}$. The saddle points reside at energy $E = -85$ meV, which is only 3 meV below the energy of the nodal point. As shown in Fig. 4a, this gives rise to an intriguing density of states with a logarithmic divergence in close proximity to a zero. Finally, below the logarithmic singularity ($E < -85$ meV), the DOS is large and almost constant, corresponding to an effective mass of about $m^* \approx 0.4 m_e$. This is an average of the almost-flat character of the lower band along $k_y - k_{\bar{Z},y} = -4.15(k_z - k_{\bar{Z},z})$ and the much steeper dispersion in the orthogonal direction.

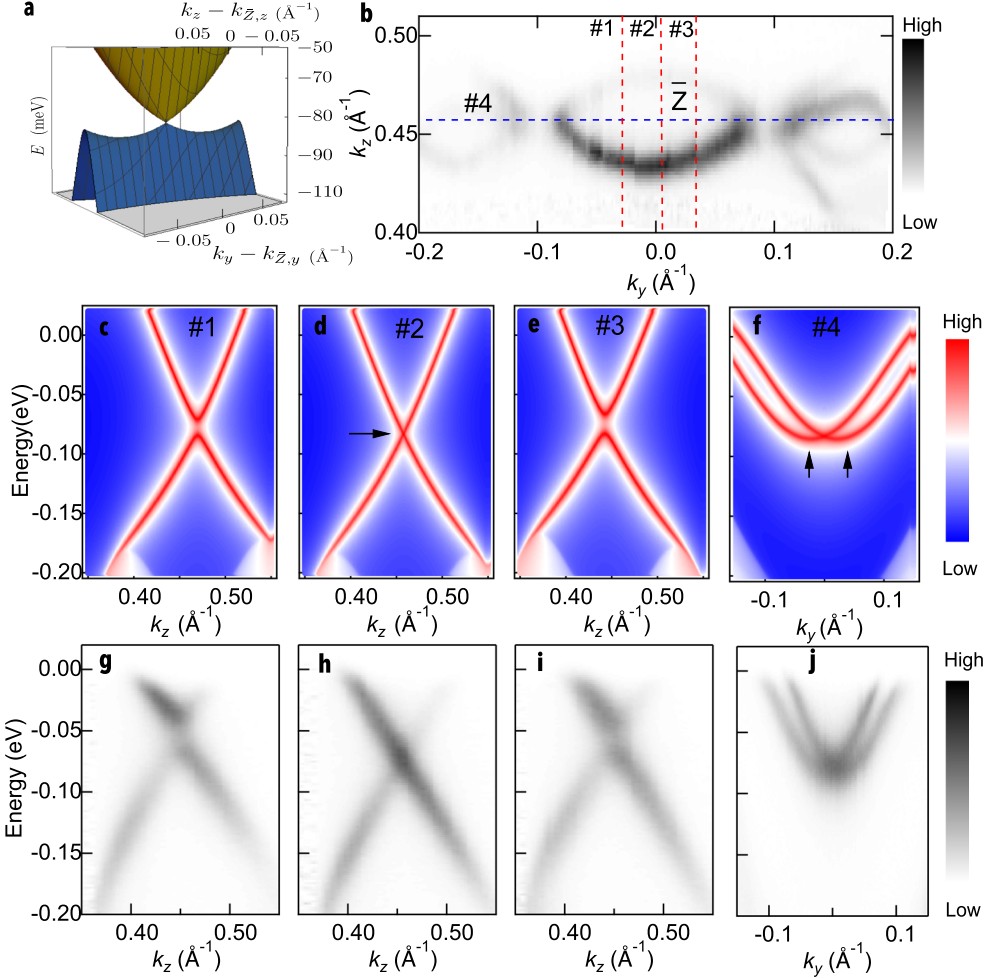

**Fig. 3 Saddle points and Dirac dispersion around the $\bar{Z}$ point. a**, $k \cdot p$ model band dispersion around the $\bar{Z}$ point, revealing a nodal point, two saddle points and the almost-flat region of the lower energy band at slightly lower energies. The two saddle points at energy $-83$ meV are responsible for the logarithmic divergence of the DOS. **b** Fermi surface plot of the ARPES intensity integrated within 10 meV of the chemical potential around the $\bar{Z}$ point. Dark areas mark location of the FS. **c**–**f** Calculated band dispersion along the vertical direction in (**b**) that is marked as red dashed lines (#1-3), respectively. **f** Calculated band dispersion along the horizontal direction in (**b**) that is marked as a blue dashed line (#4). Black arrows mark locations of band minimums. **g**–**j** ARPES data along the same cuts as in (**c**–**f**).

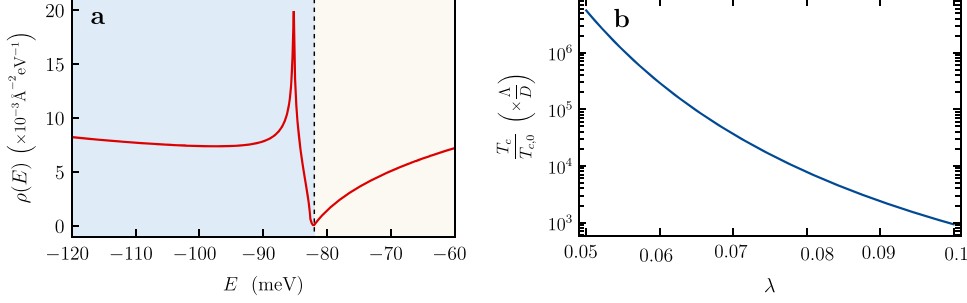

**Fig. 4 Density of states (DOS) of the effective low-energy model. a** DOS for the surface bands of $RhBi_2$ in the vicinity of the $\bar{Z}$ point as a function of energy ($E = 0$ is the location of the Fermi energy). The DOS exhibits a logarithmic divergence at $E = -85$ meV (due to saddle points) and vanishes at $E = -83$ meV (due to nodal Dirac point), and is nearly constant for energies $E < -85$ meV. **b** Enhancement of superconducting transition temperature $T_c/T_{c,0}$ at the van Hove singularity as a function of the bare dimensionless superconducting coupling $\lambda = g\rho_0/2$ in the weak-coupling regime. Here, $T_{c,0}$ is the superconducting transition temperature for chemical potential away from van Hove point, while $T_c$ is the transition temperature for $E_F$ tuned to van Hove singularity. Note that $T_c$ is normalized by $D$, while $T_{c,0}$ is normalized by the cutoff $\Lambda$.

To explore one of the possible consequences of the logarithmic divergence in the surface DOS, we calculate the enhancement of a hypothetical superconducting transition temperature $T_c$ close to the van Hove point. This provides an explicit and quantitative prediction on the expected boost of correlation effects in $RhBi_2$ arising from the surface saddle points, stimulating further experimental efforts to tune the Fermi energy close to the saddle points. Such tuning can be accomplished by chemical substitution

(e.g., Rh with Ru) or electrostatic gating. For concreteness, we consider a fully gapped $s$-wave superconducting state stabilized on the surface of RhBi$_2$. We calculate the ratio $T_c/T_{c,0}$ as the system is tuned across the van Hove singularity, where $T_{c,0}$ refers to the transition temperature away from the van Hove point. A discussion of the microscopic origin of superconductivity and possible other pairing channels is left for future work. Note that similar enhancements are expected in other interaction channels such as for density wave states. At weak coupling, $T_c$ can be calculated from the standard BCS expression

$$1 = g \int_{-\Lambda}^{\Lambda} d\xi \, \frac{\rho(\xi)}{2\xi} \tanh\left(\frac{\xi}{2T_c}\right), \qquad (4)$$

where $g$ is a momentum-independent superconducting coupling and $\Lambda$ corresponds to the Debye frequency in case of phonon-mediated superconductivity. At the saddle points the DOS diverges as $\rho(\xi) = \rho_0 \log(D/|\xi|)$, with $\rho_0 \approx 3.2 \times 10^{-3}$ Å$^{-2}$eV$^{-1}$ and $D \approx 16.5$ meV [see Fig. 4(a)]. This implies a large number of states close to $\bar{Z}$ that can pair with their time-reversed partners around $-\bar{Z}$ favoring a superconducting instability with enhanced transition temperature[37] (see also Supplementary Note 4)

$$T_c \approx 1.13 D e^{-1/\sqrt{\lambda}}. \qquad (5)$$

Here, the quantity $\lambda = g\rho_0/2 < 1$ denotes a dimensionless super-conducting coupling. Compared to the standard BCS transition temperature $T_{c,0} \approx 1.13\Lambda e^{-1/\lambda}$ for a constant DOS, $T_c$ is largely enhanced as follows from the modified dependence on the $\lambda < 1$ through $e^{-1/\sqrt{\lambda}}$ instead of $e^{-1/\lambda}$. As shown in Fig. 4(b), this results in dramatic $T_c$ enhancement over several orders of magnitude. Note that a second factor that contributes to $T_c$ enhancement is the replacement of prefactor $\Lambda$ by the inverse width of the logarithmic divergence $D$, where typically $D > \Lambda$[38].

Our results identify RhBi$_2$ as a very promising topological material that has a Dirac surface state with two saddle points close to E$_F$. DFT calculations and symmetry analysis show that triclinic RhBi$_2$ is a weak topological insulator having the simplest space group as reflected in its topological characterization. Based on the effective model, we show that the saddle point is related to a VHS. The proximity of the VHS to the Fermi level also provides opportunities to explore quantum many body instabilities to its inherent susceptibility to such instabilities. Our effective model and DFT calculations present guidance for understanding the saddle points and VHS around the Fermi level. Nevertheless the spike of DOS around 80 meV from k · p model, Experimental observation of DOS needs to be scrutinized in future studies by using different techniques such as Scanning Tunneling Microscope (STM) for several reasons. First, the photon energy and polarization of photon source may influence intensity of ARPES data. Second, the collected data always include signals from both surface and bulk states.

Apart from the discussed implications, RhBi$_2$ also hosts potential for other future pursuits. For example, the material could host intriguing new physical phenomena and new topological phase transitions when stress or strain are applied. On a related note, being a WTI in simplest form, RhBi$_2$ could also spark interest in making it a material platform to examine the role of defects. Indeed, growing a dislocation in different directions will ensure the binding of topologically protected modes when the weak index vector is parallel to the defect's Burger vector[39–42]. This pursuit is further underpinned by the unusual nature of the stacking vector (001;0) that is perpendicular to layering of the material and thus poses new relative orientations between the two, directly affecting the possibility of growing defects.

## Methods

**Sample growth.** RhBi$_2$ crystals were grown using the high temperature solution growth method out of excess Bi[43]. An initial concentration, Rh$_{20}$Bi$_{80}$, of elemental Rh and Bi was placed into a fritted alumina crucible[44] and sealed in fused silica ampoule under a partial pressure of argon. The ampoule was then heated up to 900 °C over 4 h, held there for 3 h and cooled down to 480 °C over 200 h. At this temperature the excess solution was separated from the RhBi$_2$ crystals using a centrifuge[43].The crystal structure is confirmed by X-ray diffraction (XRD) measurement.

**ARPES experiments.** Helium discharge lamp ARPES consists of a helium discharge lamp and a Scienta R8000 analyzer. The Helium discharge lamp ARPES data were collected wtih 21.2 eV photon energy. The laser ARPES system consists of a Scienta DA30 electron analyzer, picosecond Ti:Sapphire oscillator and fourth-harmonic generator[45]. Data from the laser based ARPES were collected with 6.7 eV photon energy. Angular resolution was set at ~0.1° and 1°, along and perpendicular to the direction of the analyser slit respectively, and the energy resolution was set at 2 meV. The size of the photon beam on the sample was ~30 μm in the laser system and ~0.5 mm in the helium lamp system. Samples were cleaved in-situ at a base pressure lower than $1 \times 10^{-10}$ Torr, 40 K and kept at the cleaving temperature throughout the measurement.

**Ab initio Calculations.** Band structure with spin-orbit coupling (SOC) in density functional theory[29,30] (DFT) have been calculated with PBE[46] exchange-correlation functional, a plane-wave basis set and projected augmented wave method[47] as implemented in VAS[48,49]. The experimental lattice parameters in the triclinic unit cell are used. A Monkhorst-Pack[50] ($7 \times 7 \times 7$) $k$-point mesh with a Gaussian smearing of 0.05 eV including the Γ point and a kinetic energy cutoff of 229 eV have been used. To calculate topological properties, a tight-binding model based on maximally localized Wannier functions[51–53] was constructed to reproduce closely the bulk band structure including SOC in the range of $E_F \pm 1$eV with Rh sd and Bi p orbitals. Then the spectral functions and Fermi surface of a semi-infinite RhBi$_2$ (100) surface were calculated with the surface Green's function methods[54–57] as implemented in WannierTools[58].

## Data availability

Raw data for this paper are available at Iowa State University data repository[59]: https://doi.org/10.25380/iastate.13713589. https://doi.org/10.25380/iastate.13713589.

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

## Acknowledgements

We are grateful to Thomas Iadecola for useful discussions and Sergey L. Bud'ko for help with transport measurements. This work was supported by the U.S. Department of Energy, Office of Basic Energy Sciences, Division of Materials Sciences and Engineering. R.-J.S. acknowledges funding from Trinity college, the Marie Curie programme under EC Grant agreement No. 842901 and the Winton programme at the University of Cambridge. G.F.L. acknowledges funding from the Aker Scholarship. The research (K.L., L.-L. W., B.K., T.V.T., N.H.J., P.P.O., P.C.C., and A.K.) was performed at Ames Laboratory. Ames Laboratory is operated for the U.S. Department of Energy by the Iowa State University under Contract No. DE-AC02-07CH11358. This work was also supported by the Center for Advancement of Topological Semimetals (B.K., T.V.T., N.H.J., P.P.O.), an Energy Frontier Research Center funded by the U.S. Department of Energy Office of Science, Office of Basic Energy Sciences, through the Ames Laboratory under its Contract No. DE-AC02-07CH11358.

## Author contributions

G.F.L, T.V.T., P.P.O, and R.-J.S provided theoretical modeling and interpretation. B.K. and P.C.C. designed, grew and characterized the samples. L.-L.W. performed DFT calculations. K.L., N.H.J., B.S., and A.K. performed ARPES measurements and support. The manuscript was drafted by K.L., G.F.L., L.-L.W., B.K., T.V.T., N.H.J., P.P.O., R.-J.S., P.C.C., and A.K. All authors discussed and commented on the manuscript.

## Competing interests

The authors declare no competing interests.
