## [Peer Review File · Nature Communications]

REVIEWER COMMENTS

Reviewer #1 (Remarks to the Author):

Kyungchang Lee et al., reported the discovery of both Dirac band dispersion and van Hove singularity in triclinic RhBi₂. Discovery of robust topological surface states with exotic properties could be potentially important. However, I cannot recommend the publication of this manuscript with its current form. My concerns and suggestions are the followings:

1. The manuscript lacks critical data to attribute the observed Dirac band dispersion to a topological surface state. There are many criteria for us to identify a topological surface state, such as spin-texture, absence of k_z dispersion, insensitivity to surface aging, etc. The authors made the claim based only on the consistency between ARPES and band calculation. More experimental evidences are needed.
2. The discussion of van Hove singularity is mostly limited to the well-known knowledge. It is not clear to me how the divergence of DOS could be related to the topological properties of this material. What are the differences between the van Hove singularity in normal materials and that in topological insulator? What novel phenomena do we expected to be observed if we could tune the van Hove singularities to the Fermi level?
3. The introduction and discussion contains little information. It would be good if the authors could explain the following questions: has similar anisotropic Dirac band dispersions been observed previously in other materials? Do RhBi₂ have isostructural compounds? How does RhBi₂ related to other similar compounds? Can the calculation show how to drive a topological transition in RhBi₂? How to shift the van Hove singularities to the Fermi level experimentally? What is the role of the bulk bands?

Reviewer #2 (Remarks to the Author):

Lee and coworkers studied a weak topological insulator candidate RhBi₂ using laser ARPES, DFT, and an effective $k \cdot p$ model. They found two surface Dirac points on the natural cleavage plane of the crystal, out of which one lies in close proximity (in both energy and momenta) to a van Hove singularity. They argue that this close proximity may bring novel quantum effects, and the authors provide an analytical calculation of how the VHS can significantly boost a hypothetical superconducting T_c .

Exploring the interplay between a weak TI and strong electron correlation effects is of great interest to the condensed matter community. In this regard, the theoretical and experimental works on RhBi₂ are timely and valuable. The team has obtained high quality ARPES spectra that match well with DFT predictions, though I have some questions regarding the ARPES data. My biggest concern is that it is not clear how the superconducting T_c calculation has anything to do with the Dirac point, so the authors should be more concrete about how weak TI and VHS may together lead to some novel effects. Overall, I recommend publication if the authors can revise the manuscript to address the following concerns:

1. Does the ARPES spectra show any indication of the divergence of the density of state? From the false color images, it is hard to discern any DOS enhancement from Fig. 2b and Fig. 3j, so the authors may wish to provide a DOS vs. binding energy plot next to the theoretical prediction of Fig. 4a
2. In Figure 2a, the authors may wish to explain how they obtained the orientations indicated by the

dashed line. Based on the limited Fermi surface coverage by the 6.7eV photon, it seems to me that the indicated orientation has a large uncertainty.

3. The author commented that "...Figs. 3c-e, show the lower band with negative curvature..." It looks to me the calculated lower bands are rather linear and in fact they are very slightly concave up. Where is the negative curvature?

4. When explaining the calculated spectra in Fig. 2d, the authors commented that 3D inversion symmetry is reduced to 2D rotation symmetry upon projection of all transverse momenta. However, in an experiment, at a specific photon energy, ARPES only measures a particular out-of-plane momentum. Hence, why does the Fermi surface map in Fig. 2a also look 2-fold rotational symmetric? The authors may wish to comment on which k_x momentum corresponds to the 6.7eV photon energy. Along this line, photon energy dependent spectra will be more convincing for the delegation of the surface Dirac points.

5. When calculating the enhanced hypothetical superconducting transition temperature, the authors set the Fermi level to the saddle points, which is ~ 85 meV below the actual Fermi level. This assumption seems not well justified. And as said, the entire discussion of the enhanced T_c seems to be disconnected from the fact that RhBi₂ is a weak TI.

6. There are a few minor technical points:

a. Color scales are missing in Figure 2a-f

b. What is the photon polarization for the ARPES experiment? Are the observed dispersion features dependent on the polarization?

c. In Fig. S2, the axis labels are missing and 2 should be a subscript for RhBi₂

d. Typo: "all bands are doubly degenerated in the BZ" => degenerate

e. Typo: "In cut 1 we can see the upper and lower band are separated in energy" => bands

Detailed reply to comments by the Referees:

Reviewer #1 (Remarks to the Author):

Kyungchang Lee et al., reported the discovery of both Dirac band dispersion and van Hove singularity in triclinic RhBi₂. Discovery of robust topological surface states with exotic properties could be potentially important. However, I cannot recommend the publication of this manuscript with its current form. My concerns and suggestions are the followings:

We would like to thank the Referee for carefully reading the manuscript and providing useful comments. We have taken these suggestions seriously and used them to significantly improve our manuscript.

1. The manuscript lacks critical data to attribute the observed Dirac band dispersion to a topological surface state. There are many criteria for us to identify a topological surface state, such as spin-texture, absence of k_z dispersion, insensitivity to surface aging, etc. The authors made the claim based only on the consistency between ARPES and band calculation. More experimental evidences are needed.

We thank the referee for pointing this out. Although the topological surface states are reliably identified in our DFT calculations, we agree that experimental evidence would be welcome here. We added data obtained at different photon energy showing that all features the band dispersion remain the same. We also added discussion of this topic in the revised version of the manuscript.

2. The discussion of van Hove singularity is mostly limited to the well-known knowledge. It is not clear to me how the divergence of DOS could be related to the topological properties of this material. What are the differences between the van Hove singularity in normal materials and that in topological insulator? What novel phenomena do we expected to be observed if we could tune the van Hove singularities to

the Fermi level?

As the Referee points out, van Hove singularities and associated divergences of the DOS are well-known to enhance the importance of electronic correlations and lead to interaction driven instabilities towards phases with electronic order. Well-known examples are Stoner magnetism, charge-density wave order and unconventional superconductivity. Unique about RhBi₂ is that the van Hove singularity occurs in a topological surface band. This is directly linked to nontrivial topology and low dimensionality of the material. First, the surface state is protected by nonzero weak topological index (making it more robust than trivial surface modes) and, second, the low symmetry allows for the third-order warping terms that generate saddle-point in the dispersion.

Surface van-Hove singularity potentially gives rise to exotic surface state order (e.g., superconducting), which is weakly proximity coupled into an otherwise inert bulk. Associated surface quantum criticality was theoretically predicted to realize entirely different universality classes than in the bulk, due to the mixed dimensional character of surface states [Liu, Balents, PRB **95**, 075426 (2017)]. To quantitatively estimate this scenario in RhBi₂, we calculate superconducting T_c (Fig. 4), which experiences a boost by several orders of magnitude for moderate coupling strengths. This can be done within an established weak-coupling approach. We therefore conclude that RhBi₂ is a unique system to explore this intriguing scenario experimentally for the first time.

To summarize, topology and low dimensionality are necessary to enable robust surface vH point. Correlation effects can be treated within well-established weak coupling theory and predict a boost of surface superconducting T_c , which would allow experimental study of exotic surface quantum criticality for the first time. This underpins a main point of the paper which is that RhBi₂ is a very promising material platform where symmetry [or the absence thereof] and topology meet in a viable

realization.

3. The introduction and discussion contains little information. It would be good if the authors could explain the following questions: has similar anisotropic Dirac band dispersions been observed previously in other materials? Do RhBi₂ have isostructural compounds ? How does RhBi₂ related to other similar compounds? Can the calculation show how to drive a topological transition in RhBi₂? How to shift the van Hove singularities to the Fermi level experimentally? What is the role of the bulk bands?

All these are excellent questions. We expanded the introduction in the revised version of the manuscript to address those. There are reports from other groups about anisotropic Dirac dispersion on several different topological materials such as dual topological insulator (PRB 100, 235101 (2019)) and topological crystalline insulator (PRB 92, 075131 (2015)).

We however emphasize that we are the first to introduce RhBi as a topological material platform. Given the absence of symmetry and the presence of topology by virtue of TRS we have stressed the optimal condition for topology in the manuscript. Moreover the absence of symmetry ensures the possibility of extreme warping giving the interesting van Hove singularity and according dispersion of the surface states. As made more clear now this has direct influence in terms of being prone to instabilities and other interesting physical effects. This makes RhBi a strong and yet to be fully explored material platform, the indication of which is a major merit of our work. In fact we ourselves are already exploring substitutions with other elements to address new question within this context.

In our work we focus on the surface states and the effect of electronic correlations on surface electronic order. As pointed out above, the

surface vH point is protected by nontrivial bulk topology and we expect nontrivial topology to survive the development of surface electronic order. The bulk bands are mostly inert and only experience weak correlations. As pointed out in Ref.[Liu, Balents, PRB **95**, 075426 (2017)], however, the coupling of surface to bulk leads to unusual Landau damping and exotic surface quantum criticality.

Reviewer #2 (Remarks to the Author):

Lee and coworkers studied a weak topological insulator candidate RhBi₂ using laser ARPES, DFT, and an effective k·p model. They found two surface Dirac points on the natural cleavage plane of the crystal, out of which one lies in close proximity (in both energy and momenta) to a van Hove singularity. They argue that this close proximity may bring novel quantum effects, and the authors provide an analytical calculation of how the VHS can significantly boost a hypothetical superconducting T_c.

We would like to thank the Referee for carefully reading the manuscript and very useful comments.

Exploring the interplay between a weak TI and strong electron correlation effects is of great interest to the condensed matter community. In this regard, the theoretical and experimental works on RhBi₂ are timely and valuable. The team has obtained high quality ARPES spectra that match well with DFT predictions, though I have some questions regarding the ARPES data. My biggest concern is that it is not clear how the superconducting T_c calculation has anything to do with the Dirac point, so the authors should be more concrete about how weak TI and VHS may together lead to some novel effects. Overall, I recommend publication if the authors can revise the manuscript to address the following concerns:

We thank the referee for bringing up this concern. We have addressed it also in our answers to questions 2 and 3 of Referee 1, but are happy to

discuss further here as well.

First, the presence of a nontrivial weak TI index protects the observed Dirac surface state. Importantly, the surface saddle-point (with associated vH singularity) is only allowed by the low triclinic spatial symmetry of RhBi₂. The saddle-point is the main novelty of this material that distinguishes it from other materials with protected (linearly dispersing) surface Dirac states. The importance of the saddle-point is that it implies a logarithmic enhancement of the DOS, as we have quantitatively verified using first principles theory and effective low energy modeling. While such an enhanced DOS is known to generally boost correlation effects, we included the results of a superconducting T_c calculation to provide a concrete and quantitative prediction of the expected enhancement in RhBi₂. The predicted increase of T_c by more than three orders of magnitude in the weak coupling regime demonstrates that RhBi₂ is a promising system to realize interaction-induced surface electronic order. In addition, as pointed out in Ref.[Liu, Balents, PRB **95**, 075426 (2017)], the subtle interplay of surface and bulk states may lead to exotic surface quantum critical behavior with unusual Landau damping. While theoretically appealing, such a scenario has never been experimentally observed and we think that RhBi₂ is a unique system to realize such physics, if the Fermi energy can be successfully brought to the vH point.

To address the referee's concern, we have shortened this discussion on superconducting T_c and relegated parts of the calculation to the SI and we include Ref.[Liu, Balents, PRB **95**, 075426 (2017)].

1. Does the ARPES spectra show any indication of the divergence of the density of state? From the false color images, it is hard to discern any DOS enhancement from Fig. 2b and Fig. 3j, so the authors may wish to provide a DOS vs. binding energy plot next to the theoretical prediction of Fig. 4a

Extracting DOS from ARPES data is not something that can be done reliably, nor is well accepted within the field. Two main problems are the matrix elements that can change with momentum, polarization etc., therefore different portions of the band make different contributions to integrated intensity and presence of the underlying signal from the bulk bands that is a projection along the k_z direction. We now added an explanation to this portion of the revised manuscript.

2. In Figure 2a, the authors may wish to explain how they obtained the orientations indicated by the dashed line. Based on the limited Fermi surface coverage by the 6.7eV photon, it seems to me that the indicated orientation has a large uncertainty.

Since surface Dirac points appear at TRIM points. We analyze MDCs find the orientation.

The dashed line was drawn using dispersion data (i. e. the location of the Dirac point and center of the MDC peaks at various binding energies, it is therefore quite accurate. We clarified this in the revised version of the manuscript.

3. The author commented that “...Figs. 3c-e, show the lower band with negative curvature...” It looks to me the calculated lower bands are rather linear and in fact they are very slightly concave up. Where is the negative curvature?

Thank you for pointing this out. We meant to say Fig. 3 panels c and e. In those two panels showing data away from the Dirac point, the upper and lower portion of the band are separated. Upper band has indeed positive curvature. Lower band has top at $\sim -80\text{meV}$, so its curvature is negative. Along cut #4, this band is rising up away from $k_y=0$ and thus has positive curvature. This is best illustrated in schematics showing in Fig. 3a. We now corrected the typo and clarified this portion of the discussion.

4. When explaining the calculated spectra in Fig. 2d, the authors

commented that 3D inversion symmetry is reduced to 2D rotation symmetry upon projection of all transverse momenta. However, in an experiment, at a specific photon energy, ARPES only measures a particular out-of-plane momentum. Hence, why does the Fermi surface map in Fig. 2a also look 2-fold rotational symmetric? The authors may wish to comment on which k_x momentum corresponds to the 6.7eV photon energy. Along this line, photon energy dependent spectra will be more convincing for the delegation of the surface Dirac points.

This is a very good point. The experimental data in Fig. 2a may appear to have 2-fold symmetry, but upon closer inspection it does not. The left and right surrounding of the Z pocket are quite different, this is perhaps best seen in Fig. 3b, where there is additional FS sheet merging with bulk bands to the right of Z pocket, but it is absent on the left side. This is in good agreement with calculations in Fig 2d, where the sharper surface state is present on the right side of Z pocket, but no on the left.

The ARPES data in this material are dominated by surface state and thus do not have k_x dispersion. Most of the bulk bands in calculation have broad k_x projection and result in very broad features in ARPES. This makes obtaining the exact value of k_x for 6.7 eV difficult. We can provide such an estimation with assumption of reasonable inner potential. We now added ARPES data measured at different photon energies to show that these features do not change with photon energy along with discussion of extracting the k_x values in the revised version of the manuscript.

5. When calculating the enhanced hypothetical superconducting transition temperature, the authors set the Fermi level to the saddle points, which is $\sim 85\text{meV}$ below the actual Fermi level. This assumption seems not well justified. And as said, the entire discussion of the enhanced T_c seems to be disconnected from the fact that RhBi_2 is a weak TI.

The enhancement of superconductivity occurs when the saddle point is in close proximity to the Fermi level. In principle Fermi level can be shifted by means of elemental substitution or gating and 85 meV is not out of reach for either approach. For example, substitution of Rh with Ru seems to be a very promising route. We added explanation of our hypothetical assumption and stated possible paths for achieving this in the revised version.

6. There are a few minor technical points:

a. Color scales are missing in Figure 2a-f

We thank the referee and have fixed these issues.

b. What is the photon polarization for the ARPES experiment? Are the observed dispersion features dependent on the polarization?

The polarization in our experiments is along the Gamma-Z direction. At the moment we cannot change the polarization in our setup, but most likely the ARPES intensity of all features will depend on the photon polarization. We added note about the polarization in the revised version of the manuscript.

c. In Fig. S2, the axis labels are missing and 2 should be a subscript for RhBi₂

We thank the referee and have fixed those.

d. Typo: “all bands are doubly degenerated in the BZ” => degenerate

We thank the referee and have fixed this.

e. Typo: “In cut 1 we can see the upper and lower band are separated in energy” => bands

We thank the referee and have fixed this.

REVIEWERS' COMMENTS

Reviewer #1 (Remarks to the Author):

The authors addressed most of my concerns in the revised manuscript. The existence of surface Dirac points is convincing to me with the photon energy dependent data. I would like to recommend the publication of this manuscript.

However, the authors should consider reorganizing the results and discussion parts of the manuscript. Many important discussions are included in the result part and the discussion part is more like a brief summary. It is hard for a reader to understand the significances of this manuscript and get the key information. The authors may consider showing the discovery of topological surface bands and van Hove point in the result part, while discussing all related implications, such as T_c boosting, correlation effect, unique topological phenomena, etc., in the discussion part.

Reviewer #2 (Remarks to the Author):

The authors have responded to most of my concerns. One remaining point concerns Figure 2a, where the following description is confusing: "In spite of calculated FS shows 2-fold rotational symmetry, our ARPES data shows broad projection, which may come from bulk states, on FS that is not expected on DFT calculations. This area is marked by red circle on fig 2 a." The authors may wish to adapt some of the wordings used in their response letter to explain the lack of 2-fold rotational symmetry in the measurement.

RESPONSE TO REVIEWERS' COMMENTS

Reviewer #1 (Remarks to the Author):

>The authors addressed most of my concerns in the revised manuscript. The existence of surface Dirac points is convincing to me with the photon
>energy dependent data. I would like to recommend the publication of this manuscript.

>However, the authors should consider reorganizing the results and discussion parts of the manuscript. Many important discussions are included in the
>result part and the discussion part is more like a brief summary. It is hard for a reader to understand the significances of this manuscript and get the
>key information. The authors may consider showing the discovery of topological surface bands and van Hove point in the result part, while discussing
>all related implications, such as Tc boosting, correlation effect, unique topological phenomena, etc., in the discussion part.

We would like to thank the Referee for constructive comments that significantly improved the

manuscript. Since “Discussion part is optional, we removed this heading and joined all text under “Results” section.

Reviewer #2 (Remarks to the Author):

>The authors have responded to most of my concerns. One remaining point concerns Figure 2a, where the following description is confusing: “In spite >of calculated FS shows 2-fold rotational symmetry, our ARPES data shows broad projection, which may come from bulk states, on FS that is not >expected on DFT calculations. This area is marked by red circle on fig 2 a. ” The authors may wish to adapt some of the wordings used in their >response letter to explain the lack of 2-fold rotational symmetry in the measurement.

We would like to thank the Referee for constructive comments that significantly improved the manuscript. We replaced this sentence with statement based on our previous reply as suggested by the Referee.